# Risk factors for Overweight/Obesity among people living with HIV on antiretroviral therapy: An ambidirectional cohort study at a tertiary health facility in Zambia

**Benson M. Hamooya[1], Lukundo Siame[1,2,3]\*, Matenge Mutalange[1], Chilala Cheelo[1], Kingsley Kamvuma[1], Sepiso K. Masenga[1], Chanda Chitalu[3], Sadeep Shrestha[4], Samuel Bosomprah[5]**

**1** Mulungushi University School of Medicine and Health Sciences, Livingstone, Zambia, **2** Department of Internal Medicine, Livingstone University Teaching Hospital, Livingstone, Zambia, **3** Ministry of Health, Zambia, **4** University of Alabama at Birmingham (UAB) School of Public Health, Birmingham, Alabama, United States of America, **5** Centre for Infectious Disease Research in Zambia, Zambia

\* lukundosiame23@gmail.com

## Abstract

### Background

Overweight and obesity are major concerns among people living with HIV (PLWH), particularly those on integrase inhibitors, as they elevate the risk of cardiovascular diseases. However, longitudinal data on the burden and risk factors for overweight/obesity in sub-Saharan Africa (SSA) remain limited. This study aimed to estimate the incidence and identify factors associated with overweight and obesity among PLWH who switched to a dolutegravir (DTG)-based ART regimen at Livingstone University Teaching Hospital.

### Methods

We enrolled 249 adults aged ≥18 years living with HIV on ART [non-nucleoside/nucleotide reverse transcriptase inhibitor (NNRTI) n = 174, protease inhibitor (PI) n = 21, and DTG n = 54] with a baseline body mass index (BMI) < 25 kg/m² between April 2019 and May 2020 and conducted a single follow-up assessment between December 2022 and June 2023. Participants were followed for a median of 43 months (interquartile range [IQR]: 42, 44). At follow-up, all participants were on a DTG-based regimen for a median time of 23 months (IQR: 19, 40). Demographic, clinical, and laboratory data were collected using a structured questionnaire. The primary outcome was overweight/obesity, defined as BMI ≥ 25 kg/ m². Poisson regression with robust standard errors was used to determine risk factors for being overweight and obesity.

**Data availability statement:** All relevant data are within the manuscript and its Supporting Information files

**Funding:** The author(s) received no specific funding for this work.

**Competing interests:** The authors have declared that no competing interests exist.

## Results

The median age was 44 years (interquartile range (IQR) 36, 51) at baseline, with the majority being female (59.4%, n = 148). Over a total follow-up of 871.5 person-years, 44 incident cases of overweight/obesity occurred, yielding a cumulative incidence of 17.7% (44/249) and an incidence rate of 5.05 per 100 person-years. Factors positively associated with the risk of being overweight/obesity included being married (adjusted incidence rate ratio [aIRR] 2.34; 95% CI 1.24, 4.40), lower baseline CD4 count (aIRR 4.13; 95% CI 1.41, 13.38) and higher waist circumference (WC) values (aIRR 1.07; 95% CI 1.03, 1.11). While older age was associated with a lower risk of overweight/obesity (aIRR 0.97; 95% CI 0.94, 0.99).

## Conclusion

The burden of overweight/obesity was high, and it was significantly driven by demographic, anthropometric, and immunological factors among our study participants. The findings suggest the importance of implementing targeted screening and management strategies for overweight and obesity, particularly among married individuals with higher WC values. Studies investigating the underlying mechanisms of excessive weight gain among PLWH on an integrase inhibitor-based regimen in resource-limited settings are warranted.

## Introduction

In recent years, overweight and obesity have become a major public health concern [1]. The prevalences are rising particularly in low- and middle-income countries (LMICs), particularly in the southern African region, with estimates suggesting the burden of obesity could soon rival that of infectious diseases [2,3]. Serious diseases like cardiovascular disease, diabetes, chronic kidney disease, and some cancers are associated with excessive weight gain and are leading causes of morbidity and mortality worldwide [2]. With the increased longevity of people living with HIV (PLWH) due to the benefits of antiretroviral therapy (ART), they face challenges of excessive weight gain, which has been associated with a higher risk of metabolic diseases and poorer clinical outcomes compared to HIV-negative individuals [4].

Overweight and obesity in PLWH is a complex interplay of environmental and biological factors [5]. Modern obesogenic environments, characterized by readily available high-calorie foods, smoking, alcohol consumption, sedentary lifestyles, and reduced physical activity, contribute significantly to overweight/obesity [6]. Additionally, HIV-induced inflammation and specific ART, such as those containing integrase inhibitors (INSTIs) like Dolutegravir (DTG), which are widely used in our setting, have been shown to predispose PLWH to significant weight gain [7]. Demographic factors, including age, sex, socioeconomic status, and cultural influences, also play a role in shaping individual susceptibility to overweight/obesity [8,9].

Zambia, with one of the highest HIV prevalence rates globally at 11.0%, is also experiencing a rise in obesity rates, which now affect approximately 20% of the general population, particularly women [2,10]. Despite the increasing prevalence of obesity in our setting, most studies have primarily focused on the general population and factors associated with obesity, while longitudinal studies among PLWH remain limited. This study aimed to fill this gap by investigating the incidence and risk factors associated with overweight and obesity among PLWH on ART at a tertiary hospital in the Southern Province of Zambia.

## Methods and materials

### Study design and setting

This was an ambidirectional cohort study with one follow-up assessment per participant. The median follow-up time was 43 months (IQR: 42–44), calculated as the time between baseline enrollment (April 2019–May 2020) and the follow-up visit (December 2022–June 2023). This study was conducted at Livingstone University Teaching Hospital ART clinic among adults aged 18 years or older attending routine care and ART. The hospital provides HIV care and treatment to approximately 4,000 individuals and serves as a referral center for specialized medical care in Zambia's Southern and Western provinces.

### Eligibility and sampling method

Adults (≥18 years) on ART at Livingstone University Teaching Hospital were recruited. Eligible participants had a confirmed HIV diagnosis, were on ART for ≥ 6 months and above, had a baseline BMI < 25 kg/m², and consented to participate in the study. Participants with metabolic syndrome (defined according to IDF criteria), pregnant at enrollment, or with terminal illness were excluded.

### Variables in the study

The outcome variable in this study was overweight (BMI, 25–29.9 kg/m2)/obese (BMI, ≥ 30 kg/m2), while independent variables were sociodemographic factors (age, sex, marital status, education level, and work status), behavioral factors (history of smoking, history of alcohol use, physical activity), clinical factors (blood pressure (systolic and diastolic), ART regimen, duration on ART, current duration on ART, waist circumference), and laboratory factors (viral load, CD4 count, lipid fasting profile [low-density lipoprotein (LDL), total cholesterol (TC), high-density lipoprotein (HDL)]).

### Data collection

Sociodemographic factors (age, sex, marital status, education level, and employment status), behavioral/ lifestyle (history of smoking, history of alcohol use, physical activity), clinical parameters (ART regimen, blood pressure, height, weight), and laboratory metrics (lipid profile, CD4 count, fasting glucose, viral load) were obtained directly from participants and medical records through a structured questionnaire and a data abstraction form by the trained research assistants at baseline and this was repeated at the follow up period.

Blood pressure was measured using an Omron-HEM-7120 digital equipment from the United States. After a five-minute seated rest period, blood pressure was measured three times at one-minute intervals using an Omron-HEM-7120 digital monitor. The average of the three readings was then calculated to determine each participant's blood pressure. A height measurement chart, a digital scale, and a tape measure were used to determine height, weight, and waist circumference, respectively.

The ART regimens were retrieved from the electronic medical record (SmartCare) and were classified as follows: A combination of integrase strand transfer inhibitors (INSTIs) that includes dolutegravir (DTG) and tenofovir disoproxil fumarate/lamivudine (TDF/3TC) or tenofovir alafenamide/lamivudine (TAF/3TC). Non-nucleoside reverse transcriptase inhibitor (NNRTI) regimens included either efavirenz (EFV) or nevirapine (NVP) in combination with one of the following nucleoside reverse

transcriptase inhibitors (NRTIs): abacavir and lamivudine/emtricitabine (ABC/XTC) or tenofovir disoproxil fumarate and lami-vudine/emtricitabine (TDF/XTC). The protease inhibitor (PI) regimens included either lopinavir/ritonavir (LPV/r) or atazanavir/ritonavir (ATV/r) in combination with one of the NRTI combinations: ABC/XTC, zidovudine/XTC (AZT/XTC), or TDF/XTC.

### Definitions

Body mass index (BMI) was categorized as underweight (<18.5 kg/m²), normal weight (18.5–24.9 kg/m²), overweight (25.0–29.9 kg/m²), or obese (≥30 kg/m²) [11]

A participant was classified as physically active if, during a typical week, they engaged in activities such as carrying or lifting heavy loads, digging, crushing stones, or construction work for at least 10 continuous minutes, or participated in moderate to vigorous-intensity sports, fitness, or recreational activities like running, football, cycling, swimming, or volley-ball for at least 10 continuous minutes.

### Data analysis

Data were entered into Microsoft Excel 2013 for cleaning. Statistical analysis was conducted using Stata version 15 (Stata Corporation, College Station, TX, USA). Categorical variables were summarized using frequencies and percentages. We used the Shapiro-Wilk test and Q-Q plots for continuous variables to determine the normality of the data; then, medians and inter-quartile ranges were used to summarize the data if it wasn't normally distributed. The chi-square test was used to examine the statistical significance between two categorical variables. The Wilcoxon rank sum test was used to determine the statistical significance between the two medians. Robust Poisson logistic regression was employed to calculate the incidence rate ratios (IRR) and 95% confidence intervals (CI) for the associations between overweight/obesity and other study covariates. Poisson regression with robust standard errors is the recommended method for estimating risk ratios (RR) in cohort studies when the incidence of outcome is common (i.e., > 10%), as it provides more accurate estimates compared to logistic regression, which tends to overestimate relative risks or risk ratios [12,13]. The baseline variables included in the final regression model were chosen based on evidence from previous studies and their significance in bivariable analysis. To assess multicollinearity among predictors, the Variance Inflation Factor was used. Statistical significance was defined as p < 0.05.

### Ethics

The study received ethical approval from the Mulungushi University School of Medicine and Health Sciences Research Ethics Committee (MUSoMHS-REC- Ref. No: SMHS-MU3-2022-12) valid from 17th June 2022–17th June 2023 and data collection was done from 8th December 2022–17th January 2023. Participants signed informed consent after being informed of the study's purpose in a language they could understand. The study data were fully anonymized throughout both collection and processing stages, with all personally identifiable information removed. In line with best practices for research transparency, we strictly followed the Strengthening the Reporting of Observational Studies in Epidemiology (STROBE) reporting guidelines (S1 File).

### Results

Of the 512 participants screened between April 2019 and May 2020, 200 were excluded (124 with metabolic syndrome, 76 overweight/obese). Among the remaining 312, 63 were excluded during follow-up (1 death, 36 non-response, 1 preg-nancy, 19 transfers, 6 lost to follow-up), leaving 249 for analysis (Fig 1).

### Basic demographic and clinical characteristics at baseline

The study included 249 participants with a median age of 44 years (interquartile range (IQR) 36, 51), and the majority were female (59.4%, n = 148). Most of the participants were not married (53.1%, n = 132). A higher proportion of the par-ticipants had secondary education (55.4%, n = 138). Most participants were self-employed (47.4%, n = 118). The median

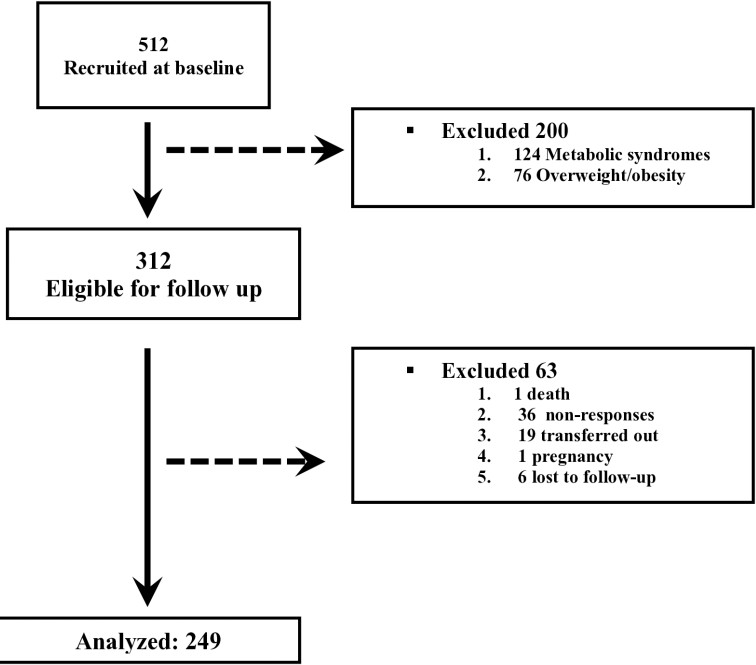

**Fig 1. Flowchart of screened and eligible participants.**

weight of the participant was 57 kg (IQR: 51, 62). At baseline, 69.9% (n = 174) were on NNRTI, 8.4% (n = 21) on PI, and 21.7% (n = 54) on DTG, while at the follow-up period, all participants were switched to a DTG-based regimen (see supplementary **Table 1** in S3 File). At baseline, the overall median duration on ART was 102 months (IQR: 60–144) (see Table 1) increasing to 155 months (IQR: 97–189) at follow-up (see supplementary **Table 1** in S3 File). The median duration on the current DTG-based regimen at follow-up was 23 months (IQR: 19–40) (see supplementary **Table 1** in S3 File). Median values of waist circumference was 76 cm (IQR: 71, 80). Total cholesterol of the participant was 4.4 mmol/L (IQR: 3.5, 5.2) and HDL level was 1.3 mmol/L (IQR: 1.1, 1.7) (**see** Table 1).

### Body mass index (BMI) changes from baseline to end of follow-up

Among the 60 participants who were underweight at baseline, 32 (53.3%) remained underweight, 23 (38.3%) transitioned to a normal weight, 3 (5.0%) became overweight, and 2 (3.3%) moved into the obese category. Of the 189 participants with a normal weight at baseline, 11 (5.8%) shifted to underweight, 139 (73.5%) maintained a normal weight, 36 (19.1%) became overweight, and 3 (1.6%) progressed to obesity (**see** Table 2).

### Incidence of overweight/obesity among the study participants

Over a total follow-up of 871.5 person-years (median: 42 months; IQR: 40–44), 44 incident cases of overweight/obesity were recorded, corresponding to a cumulative incidence of 17.7% (44/249), with 15.7% (39/249) classified as overweight and 2.0% (5/249) as obese. This resulted in an incidence rate of 5.05 per 100 person-years.

### Relationship of overweight/obesity with other study variables

Participants who were not married had a higher proportion of being overweight or obese compared to married participants (68.2% vs. 31.8%). A higher proportion of individuals who were overweight or obese had a higher BMI at baseline

**Table 1. Baseline demographic and clinical characteristics of participants. (N = 249).**

| Variable | |
|---|---|
| **Age, *years (IQR)*** | **44 (36, 51)** |
| **Sex, *n (%)*** | |
| Male | 101 (40.6) |
| Female | 148 (59.4) |
| **Marital status, *n (%)*** | |
| Married | 117 (46.9) |
| Unmarried | 132 (53.1) |
| **Education level, *n (%)*** | |
| No formal schooling | 2 (0.8) |
| Primary | 74 (29.7) |
| Secondary | 138 (55.4) |
| Tertiary | 35 (14.1) |
| **Work status, *n (%)*** | |
| Government employee | 18 (7.2) |
| Non-government | 45 (18.1) |
| Self-employed | 118 (47.4) |
| Unemployed | 68 (27.3) |
| **Smoking status, *n (%)*** | |
| No | 234 (94.3) |
| Yes | 14 (5.7) |
| **Systolic BP, *mmHg (IQR)*** | **118 (108.7, 132)** |
| **Diastolic BP, *mmHg (IQR)*** | **76 (68, 82)** |
| **Weight, *kg (IQR)*** | **57 (51, 62)** |
| **Height, *cm (IQR)*** | **165.5 (160, 172)** |
| **Body mass index, *n (%)*** | |
| Underweight | 60 (24.1) |
| Normal | 189 (75.9) |
| Overweight | |
| Obese | |
| **ART regimen at baseline, *n (%)*** | |
| NNRTI (EFV/NVP) | 174 (69.9) |
| PI (LPV/r, ATV/r) | 21 (8.4) |
| INSTI (DTG) | 54 (21.7) |
| **Baseline NRTI, *n (%)*** | |
| ABC/3TC | 7 (2.8) |
| AZT/3TC | 16 (6.4) |
| TDF/3TC | 226 (90.8) |
| TAF/3TC | |
| **Duration on ART, *months (IQR)*** | 102 (60, 144) |
| **Current duration on ART, *months (IQR)*** | 67 (9, 71) |
| **Viral load, *copies/mL*** | 20 (0, 371) |
| **CD4 count, *cells/μL (IQR)*** | 510 (365, 772) |
| **Waist circumference, *cm (IQR)*** | 76 (71, 80) |
| **Hip circumference, *cm (IQR)*** | 89 (84, 94) |
| **LDL cholesterol, *mmol/L (IQR)*** | 1.9 (1.3, 2.4) |
| **Total cholesterol, *mmol/L (IQR)*** | 4.4 (3.5, 5.2) |

*(Continued)*

**Table 1.** (Continued)

| Variable | |
|---|---|
| HDL cholesterol, *mmol/L (IQR)* | 1.3 (1.1, 1.7) |
| Physically active, *n (%)* | |
| No | 175 (72.3) |
| Yes | 67 (27.7) |

Note: Values are expressed as median (IQR) for continuous variables and n (%) for categorical variables.

Abbreviations: BP = Blood Pressure; ART = Antiretroviral Therapy; NNRTI = Non-Nucleoside Reverse Transcriptase Inhibitor; PI = Protease Inhibitor; INSTI = Integrase Strand Transfer Inhibitor; NRTI = Nucleoside Reverse Transcriptase Inhibitor; EFV = Efavirenz; NVP = Nevirapine; LPV/r = Lopinavir/ritonavir; ATV/r = Atazanavir/ritonavir; DTG = Dolutegravir; ABC = Abacavir; 3TC = Lamivudine; AZT = Zidovudine; TDF = Tenofovir Disoproxil Fumarate; TAF = Tenofovir Alafenamide; LDL = Low-Density Lipoprotein; HDL = High-Density Lipoprotein; IQR = Interquartile range

**Table 2. Changes in BMI from baseline to end of follow-up.**

| | | BMI at 4 years | | | |
|---|---|---|---|---|---|
| | Baseline BMI n (%) | Underweight n (%) | Normal n (%) | Overweight n (%) | Obese n (%) |
| Underweight | 60(24.1) | 32(53.3) | 23(38.3) | 3(5.0) | 2(3.3) |
| Normal | 189(75.9) | 11(5.8) | 139(73.5) | 36(19.1) | 3(1.6) |
| Total | 249 (100) | | | | |

compared to those without (22.9 kg/m2 vs. 20.0 kg/m2, p < 0.001). Median waist circumference was higher in the overweight/obese group compared to individuals who were not overweight or obese (79.5 cm vs. 74 cm, p < 0.001) (**see** Table 3).

### Regression analysis of the factors associated with overweight/obesity.

At bivariable analysis, married participants had a significantly 90% increased risk of overweight/obesity compared to unmarried individuals, incidence rate ratio (IRR) 1.90; 95% confidence interval (CI) 1.06, 3.41. A unit increase in waist circumference was significantly associated with a 6% increased risk of being overweight/obese, IRR 1.06: 95%CI 1.03, 1.09 (**see** Table 4).

 At multivariable analysis, a one-year increase in age was significantly associated with a 3% reduced risk of being overweight/obese, IRR 0.97: 95%CI 0.94, 0.99. The married participants had a 2.34 times higher risk of being overweight/obese compared to unmarried individuals. A unit increase in waist circumference was significantly associated with a 7% increased risk of overweight/obesity, IRR 1.07; 95%CI 1.24, 4.40. Individuals with a CD4 count below 200 cells/µL at baseline were 4.13 times more likely to be overweight/obese compared to those with a CD4 count above 500 cells/µL (see Table 4).

## Discussion

This study aimed to explore the incidence, and the risk factors associated with overweight and obesity in PLWH. After a median follow up period of 42 months, the cumulative incidence (17%) of overweight or obese people was high among the study participants. The development of overweight/obesity was positively associated with being married, increasing waist circumference, and having a CD4 count below 200 cells/µl, while older age appeared to have a protective effect.

 The cumulative incidence of overweight/obesity in this study was lower than in Tanzania (2018) at 35% [14]. The observed differences may stem primarily from the longer follow-up period in Tanzania, which was 10 years compared to

**Table 3. Relationship Between Overweight/Obesity and other study variable.**

| Variable | Overweight/Obese (n = 44) | Not Overweight/Obese (n = 205) | P-value |
|---|---|---|---|
| **Age**, *years* | 43.5 (37.5, 49.5) | 44.0 (36.0, 51.0) | 0.985 |
| **Sex** | | | 0.125 |
| Male | 14 (31.8%) | 91 (44.4%) | |
| Female | 30 (68.2%) | 114 (55.6%) | |
| **Marital status** | | | **0.026** |
| Married | 14 (31.8%) | 103 (50.2%) | |
| Unmarried | 30 (68.2%) | 102 (49.8%) | |
| **Education level** | | | 0.282 |
| No formal schooling | 0 (0.0%) | 2 (1.0%) | |
| Primary | 10 (22.7%) | 64 (31.2%) | |
| Secondary | 30 (68.2%) | 108 (52.7%) | |
| Tertiary | 4 (9.1%) | 31 (15.1%) | |
| Work status | | | 0.768 |
| **Government-employed** | 3 (6.8%) | 15 (7.3%) | |
| Non-government | 7 (15.9%) | 38 (18.5%) | |
| Self-employed | 24 (54.6%) | 94 (45.9%) | |
| Unemployed | 10 (22.7%) | 58 (28.3%) | |
| **Smoking** | | | 0.285 |
| No | 43 (97.7%) | 191 (93.6%) | |
| Yes | 1 (2.3%) | 13 (6.4%) | |
| **Alcohol use** | | | 0.712 |
| No | 20 (47.6%) | 102 (50.8%) | |
| Yes | 22 (52.4%) | 99 (49.2%) | |
| **Systolic BP, mmHg** | 115 (110,131.2) | 118 (108.3,133.0) | 0.745 |
| **Diastolic BP, mmHg** | 75.7 (69.3, 81.5) | 76.3 (68.0,82.0) | 0.853 |
| **BMI, kg/m²** | 22.9 (20.8,23.9) | 20.0 (18.2,22.2) | **<0.001** |
| **ART regimen** | | | 0.287 |
| NNRTI (EFV/NVP) | 35 (79.6%) | 139 (67.8%) | |
| PI (LPV/r or ATV/r) | 2 (4.5%) | 19 (9.3%) | |
| INSTI (DTG) | 7 (15.9%) | 47 (22.9%) | |
| **Baseline NRTI** | | | 0.382 |
| ABC/3TC | 0 (0.0%) | 7 (3.4%) | |
| AZT/3TC | 2 (4.6%) | 14 (6.8%) | |
| TDF/3TC | 42 (95.5%) | 184 (89.8%) | |
| **Duration on ART, days** | 27 (0,644) | 20 (0, 289.5) | 0.241 |
| **Waist circumference, cm** | 79.5 (76.0, 84.0) | 74.0 (70.0,79.0) | **<0.001** |
| **Viral load, copies/mL** | | | 0.269 |
| <200 | 27 (61.4%) | 147 (71.7%) | |
| 200–1000 | 8 (18.1%) | 21 (10.2%) | |
| ≥1000 | 9 (20.5%) | 37 (18.1%) | |
| CD4 count, cells/µL | | | 0.156 |
| <200 | 6 (13.7%) | 13 (6.3%) | |
| 200–500 | 14 (31.8%) | 88 (42.9%) | |
| >500 | 24 (54.6%) | 104 (50.7%) | |

*(Continued)*

**Table 3.** (Continued)

| Variable | Overweight/Obese (n = 44) | Not Overweight/Obese (n = 205) | P-value |
|---|---|---|---|
| **LDL**, mmol/L | 2.1 (1.3,2.6) | 1.9 (1.3, 2.4) | 0.317 |
| **Total cholesterol,** mmol/L | 4.5 (3.7,5.4) | 4.3 (3.5, 5.1) | 0.651 |
| **HDL**, mmol/L | 1.3 (1.1,1.7) | 1.3 (1.1, 1.7) | 0.656 |
| **Physically active** | | | 0.537 |
| No | 32 (76.2%) | 143 (71.5%) | |
| Yes | 10 (23.8%) | 57 (28.5%) | |

Note: N total observation for the variable, ** variable presented as median (lower quartile, upper quartile),*variable presented as frequency (percentage). Bold p-values indicate statistical significance (p < 0.05). Abbreviations: BMI – body mass index; ART – antiretroviral therapy; NNRTI – non-nucleoside reverse transcriptase inhibitor; PI – protease inhibitor; INSTI – integrase strand transfer inhibitor; NRTI – nucleoside reverse transcriptase inhibitor; EFV – efavirenz; NVP – nevirapine; LPV/r – lopinavir/ritonavir; ATV/r – atazanavir/ritonavir; DTG – dolutegravir; TDF – tenofovir disoproxil fumarate; ABC – abacavir; AZT – zidovudine; 3TC – lamivudine; HDL – high-density lipoprotein; LDL – low-density lipoprotein.

our 4 years, and differences in ART status, as the Tanzanian population was mostly ART-naïve and younger, while ours involved ART-experienced individuals and older. The incidence observed in our setting, primarily among this urban population, is a public health concern due to the higher risk of cardiometabolic conditions among PLWH. This incidence may be caused by urbanization and changes in the built environment, which have reduced opportunities for physical activity [12,13]. Concurrently, a transitional nutritional shift towards highly processed, energy-dense foods has occurred, fueled by their increased availability, affordability, and accessibility due to the rise of supermarkets and aggressive marketing strategies in our setting [16]. Furthermore, the stigma surrounding HIV still prevalent in our setting, which often leads to the perception that being overweight is a sign of good health, this may drive some PLWH to gain weight as part of a 'return to health,' potentially contributing to excessive weight gain [2,15,16].

This study suggests that contrary to the generally observed trend of increased obesity with age, a one-year increase in age is associated with a slight reduction in the risk of being overweight or obese. This finding contradicts previous research that has linked aging with a higher likelihood of obesity [9,17]. This finding requires mechanistic studies to understand the relationship being age and obesity among PLWH in our setting.

The incidence of overweight/obesity was significantly higher in married participants compared to unmarried individuals in this current study. This is consistent with a study in Tanzania (2016) [17]. Overweight and obesity are more common among married adults and stable relationship, particularly among married men, because household food distribution normally favors men and men are more inclined to eat out frequently [17–19]. Women, on the other hand, are more inclined to take care of themselves whether or not they are married, but marriage has a greater impact on men [17,20].

In the current study, Individuals with a low CD4 count at baseline have a significantly higher risk of being overweight/obesity compared to those with a higher CD4 count. This study aligns with several studies that have reported similar results, however the reason to this association remains fluid [14,21–23]. The return to health in advanced HIV appears dysregulated by INSTIs, converting immune reconstitution into a driver of excessive adiposity [14,24]. This process occurs in two phases: an initial rapid phase of tissue repair followed by a slower, sustained phase as immune function improves. Individuals with significant LBM deficits at the onset of recovery tend to experience more pronounced weight gain, which can include fat accumulation, ultimately increasing their risk of being overweight and obesity [14,24].

Waist circumference showed a significant correlation with overweight/obesity, consistent with findings from previous studies [25,26]. Waist circumference, combined with BMI, provides a better prediction of health risks compared to BMI alone [27]. It effectively identifies central obesity, which is associated with cardiometabolic risk factors such as hypertension, dyslipidemia, and hyperglycemia, all of which are linked to adverse outcomes [27]. Thus, routine measurement of

**Table 4. Association between baseline sociodemographic, lifestyle and clinical factors and incidence of overweight/obesity.**

| Variable | Bivariable analysis | | Multivariable analysis | |
|---|---|---|---|---|
| | IRR (95%CI) | P-value | IRR (95%CI) | P-value |
| Age | 0.99 (0.98, 1.01) | 0.589 | 0.97 (0.94, 0.99) | **0.018** |
| **Sex** | | | | |
| Male | Ref | | Ref | |
| Female | 1.57 (0.87, 5.53) | 0.134 | 2.23 (0.90, 5.44) | 0.083 |
| **Marital status** | | | | |
| Unmarried | Ref | | Ref | |
| Married | 1.90 (1.06, 3.41) | **0.032** | 2.34 (1.24, 4.40) | **0.008** |
| **Education level** | | | | |
| Primary | Ref | | | |
| Secondary | 1.65 (0.85, 3.20) | 0.136 | Ref | |
| Tertiary | 0.87 (0.29, 2.58) | 0.8 | 1.34 (0.69, 2.62) | 0.379 |
| **Work status** | | | 0.48 (0.11,1.99) | 0.314 |
| Government employee | Ref | | Ref | |
| Non-government | 0.93 (0.27, 3.22) | 0.913 | 0.91 (0.24, 3.44) | 0.891 |
| Self-employed | 1.22 (0.41, 3.65) | 0.722 | 0.86 (0.25, 2.88) | 0.801 |
| Unemployed | 0.88 (0.27, 2.88) | 0.836 | 0.67 (0.19, 2.34) | 0.532 |
| **Smoke** | | | | |
| No | Ref | | Ref | |
| Yes | 0.84 (0.28, 2.47) | | 0.95 (0.13, 7.07) | 0.966 |
| **Alcohol** | | | | |
| No | Ref | | Ref | |
| Yes | 1.31 (0.75, 2.27) | | 1.26 (0.69, 2.31) | 0.459 |
| **ART** | | | | |
| NNRTI (EFV & NVP) | Ref | | Ref | |
| PI (LPV/r & ATZ/r) | 0.47 (0.12, 1.83) | 0.279 | 0.55 (0.18,1.72) | 0.312 |
| INSTI (DTG) | 0.64 (0.30, 1.37) | 0.253 | 0.70 (0.29, 1.71) | 0.431 |
| **Median current duration on ART in months** | 0.99 (0.98, 1.01) | 0.871 | 1.01 (0.99, 1.02) | 0.145 |
| **waist circumference** | 1.06 (1.03, 1.09) | **<0.001** | 1.07 (1.03,1.11) | **< 0.001** |
| **CD4 count, cells/µL** | | | | |
| > 500 | Ref | | Ref | |
| < 200 | 1.68 (0.79, 3.58) | 0.176 | 4.13 (1.41, 13.38) | **0.010** |
| 200- 500 | 0.73 (0.39, 1.34) | 0.314 | 1.07 (0.55, 2.11) | 0.826 |
| **LDL**, mmol/L | 1.02 (0.77, 1.33) | 0.911 | 0.95 (0.68, 1.34) | 0.788 |
| **Total cholesterol**, mmol/L | 1.2 (0.89, 1.29) | 0.453 | 1.21 (0.78,1.88) | 0.404 |
| **HDL**, mmol/L | 1.02 (0.67, 1.55) | 0.935 | 0.98 (0.55, 1.74) | 0.945 |
| **Lymphocytes**, (109 cells/l) | 0.98 (0.63, 1.54) | 0.931 | 1.08 (0.63, 1.84) | 0.774 |
| **Physically Active** | | | | |
| No | Ref | | Ref | |
| Yes | 0.90 (0.52,1.54) | 0.697 | 0.99 (0.54, 1.82) | 0.976 |

Note: IRR incidence rate ratio, Ref reference group, ART- antiretroviral therapy, NNRTI non-nucleoside/nucleotide reverse transcriptase inhibitor (EFV = efavirenz and NVP = Nevirapine), PI protease inhibitor (LPV/r = lopinavir/ritonavir and ATV/r = atazanavir/ritonavir), INSTI integrase strand transfer inhibitor (DTG = dolutegravir), NRTI nucleotide reverse transcriptase inhibitor, TDF/3TC tenofovir disoproxil fumarate/lamivudine, ABC/3TC abacavir/lamivudine, AZT/3TC zidovudine/lamivudine, LDL-c Low-Density Lipoprotein Cholesterol, HDL High-Density Lipoprotein, bold p < 0.05

waist circumference should be incorporated into HIV care, and health practitioners should be trained to properly perform this measurement [27].

In the current study, the DTG-based regimen among this cohort was not associated with an increased risk of obesity. This aligns with the findings of Mounzer et al. (2021) and Guaraldi et al. (2021) [28,29].However, other studies have demonstrated a positive association between DTG use and obesity, particularly among individuals receiving TAF-containing backbones unlike in our studies where all participants had shifted to TDF [20,30–32]. Other reason for this finding in our study could be attributed to the fact related to variations in the impact of DTG on weight gain, which could depend on factors such as sex, baseline BMI, CD4 count, and tuberculosis coinfection [30]. Additionally, all the participants by the end of the follow-up period were on a DTG-based regimen, and there is a high probability that it impacted on the excessive weight gain among our study participants. Nevertheless, further long-term studies are needed to fully understand the risk of a DTG-based regimen with TDF, which is the main treatment therapy in our setting.

The study has some strengths and weaknesses. The study has included only a single follow-up period, making it impossible to capture short-term variability in weight, potentially masking transient weight fluctuations. Future studies with serial measurements could better elucidate the trajectory and timing of weight gain in this population. Information on diet and/or food intake was not included. The reliance on self-reported variables, such as physical activity, alcohol consumption, and smoking, introduces the potential for recall bias. Despite these weaknesses, the study is one of the few longitudinal studies that have explored the incidence of overweight/obesity and risk factors among PLWH in the era of integrase inhibitors. The study provides valuable information concerning the longitudinal burden of overweight/obesity and risk factors among PLWH, information imperative in designing public health actions.

## Conclusion

This study found a high incidence of overweight and obesity among PLWH on ART. Being married, low CD4 count, and increased waist circumference were positively associated with overweight/obesity, while age was inversely associated. These findings highlight the need for targeted weight management within HIV care, especially for high-risk groups. Longitudinal studies with serial measurements are needed to clarify the impact and timing of weight gain from integrase inhibitors in this population.

## Supporting information

**S1 File. STROBE checklist.**
(DOCX)

**S2 File. Dataset.**
(XLSX)

**S3 File. Supplementary Table.**
(DOCX)

## Acknowledgments

We would like to extend our gratitude to the HAND group, chaired by Professor Masenga and Dr. Hamooya, for their invaluable support, as well as to Mulungushi University for their continued support.

## Author contributions

**Conceptualization:** Benson M. Hamooya, Sadeep Shrestha, Samuel Bosomprah.

**Data curation:** Benson M. Hamooya, Lukundo Siame, Samuel Bosomprah.

**Formal analysis:** Benson M. Hamooya, Lukundo Siame, Sepiso K. Masenga, Samuel Bosomprah.

**Funding acquisition:** Benson M. Hamooya, Samuel Bosomprah.

**Investigation:** Benson M. Hamooya, Samuel Bosomprah.

**Methodology:** Benson M. Hamooya, Samuel Bosomprah.

**Project administration:** Benson M. Hamooya.

**Resources:** Benson M. Hamooya.

**Software:** Benson M. Hamooya.

**Supervision:** Benson M. Hamooya, Lukundo Siame, Chilala Cheelo, Sepiso K. Masenga, Chanda Chitalu, Sadeep Shrestha, Samuel Bosomprah.

**Validation:** Benson M. Hamooya, Lukundo Siame, Matenge Mutalange, Chilala Cheelo, Kingsley Kamvuma, Sepiso K. Masenga, Chanda Chitalu, Sadeep Shrestha, Samuel Bosomprah.

**Visualization:** Benson M. Hamooya, Lukundo Siame, Matenge Mutalange, Chilala Cheelo, Kingsley Kamvuma, Sepiso K. Masenga, Chanda Chitalu, Sadeep Shrestha, Samuel Bosomprah.

**Writing – original draft:** Benson M. Hamooya, Lukundo Siame, Matenge Mutalange, Sepiso K. Masenga, Chanda Chitalu, Sadeep Shrestha, Samuel Bosomprah.

**Writing – review & editing:** Benson M. Hamooya, Lukundo Siame, Matenge Mutalange, Chilala Cheelo, Kingsley Kamvuma, Sepiso K. Masenga, Chanda Chitalu, Samuel Bosomprah.

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
