## [Decision Letter · Decision Letter 0]

9 Jun 2025

Dear Dr. Siame,

Thank you for submitting your manuscript to PLOS ONE. After careful consideration, we feel that it has merit but does not fully meet PLOS ONE’s publication criteria as it currently stands. Therefore, we invite you to submit a revised version of the manuscript that addresses the points raised during the review process.

We look forward to receiving your revised manuscript.

Kind regards,

Anne Kapaata

Academic Editor

PLOS ONE

Additional Editor Comments (if provided):

Reviewers' comments:

Reviewer's Responses to Questions

**Comments to the Author**

1. Is the manuscript technically sound, and do the data support the conclusions?

Reviewer #1: Yes

Reviewer #2: No

2. Has the statistical analysis been performed appropriately and rigorously?

Reviewer #1: Yes

Reviewer #2: No

3. Have the authors made all data underlying the findings in their manuscript fully available?

Reviewer #1: Yes

Reviewer #2: Yes

4. Is the manuscript presented in an intelligible fashion and written in standard English?

Reviewer #1: Yes

Reviewer #2: No

Reviewer #1: 1. The authors should use a consistent nosology for people living with HIV (PLWH); there are two nosologies used eg People living with HIV and people with HIV

2. The findings should be stated and then referenced with figues/table and not the other way round.

3. The conclusion did not explicitly answer the objective

Reviewer #2: The methods and statistical analysis and results sections are not clear. It is not clear how long the follow-up period was, and what the interval of follow-up for each participant was. Also, it is not clear what was done at the different follow-up periods. It is not clear what the final sample size is, since the authors indicate they recruited 388 participants and included 249, but after removing all the individuals excluded, the remaining number is less than 190, and not 249. The comparison of the baseline characteristics between baseline and follow-up is quite unusual and should be removed. The tables need to be edited in line with the journal recommendations. In the write-up on 'univariate analysis, which should be bivariate analysis, he needs to include all the variables that were entered in the multivariate analysis. The authors indicate that after checking for normality, they found data not to be normally distributed, why then do they use the chi-square and the poisson regression for non-normally distributed data?

**Do you want your identity to be public for this peer review?** For information about this choice, including consent withdrawal, please see our Privacy Policy

Reviewer #1: **Yes: ** Sody Munsaka

Reviewer #2: No

---

## [Author Response · Author response to Decision Letter 1]

24 Jun 2025

24 /06/2025

PLOS ONE Journal

Dear Editor,

Ref: Submission of a revised research article for peer review and publication consideration

Reference to the above-mentioned subject. I am writing to submit a revised original research article titled “Risk Factors for Overweight/Obesity among People living with HIV on Antiretroviral Therapy: A cohort study at a Tertiary Health Facility in Zambia”.

Academic Editor comments

Major comments to address

1. Author needs to clearly state the total duration of follow up and what was the interval of follow up per patient because as is it is not clear whether follow up was 2 or 4 years

Response: Thank you very much. We have made it clear now in the abstract

2. For cohort studies, its always good to report the follow-up time in terms of person-years and this information is missing.

Response: Thank you very much. We have now reported the follow-up in terms of person-years

3. Under eligibility and sampling method, the numbers don’t add up, how many people were screened, how many were excluded per reason of exclusion and how many made it to the final analysis? It’s stated that screened 388 and recruited 249….and excludeded most people due to overwight, death, non-response,pregnancy, transfer to another facility and lost to follow up ….from this section it looks like only 49 participants went into the final study study?

Response: Thank you very much. We have now rectified this problem to align with the fin 249 which was followed up.

4. The reasons for exclusion belong to the result section and not materials and methods. Author refers to figure 1, however there is no figure 1 in the write up.

Response: Thank you very much. We have moved the reason for exclusion to the result section, and we have uploaded Figure 1 as a separate file.

5. The statement on measurement of blood pressure is not clear, was the waiting for one minute of five minutes.

Response: Thank you very much. We have now clarified this statement

6. If the Shapiro-wilk test showed that the data were not normally distributed, why did the authors then use chi-square tests and the poisson regression which are for normally distributed data

Response: Thank you very much for the question. The Shapiro-Wilk test was used to see whether the data measured on a ratio/interval scale (quantitative data) followed a normal distribution. The chi-square test was used to determine a statistical relationship between two categorical variables. The Poisson regression with robust standard errors was used to estimate the factors (in which rate ratios where generated) associated with overweight/obesity (binary outcome).

7. Under data analysis, authors mention that “variables included in the model were chosed based on evidence from previous studies” does this mean that there was no bivariate analysis and checking for confounding and interaction terms?

Response: Thank you very much. The final model was based on literature and those variables that were significant in the bivariate analysis, which we have now clarified.

8. Under results authors mention basic demographics and clinical characteristics at baseline and follow-up…for cohort studies should the results not be reported at baseline and end of follow up?

Response: Thank you very much for the suggestion. We have corrected this now as suggested.

9. At what point during follow up were the partcipants switch to DTG? And was this taken into consideration during analysis?

Response: Thank you very much. We collect data on how long each participant was on DTG, and we have indicated the information in the manuscript in terms of the median time participants were on a DTG-based regimen “The median duration on the current DTG-based regimen at follow-up was 23 months (IQR: 19–40)”.

10. The median duration on ART at baseline was 102 (60, 144) months…is this the Interquartile range or minimum maximum time?

Response: Thank you for its interquartile range, as we have defined in the table keys and in the narration section of the results.

11. Edit table 1 according to joural specifications….some cells are empty, and the N seems to be in the wrong columns.

Response: Thank you for your observation. We have edited the table according to journal specifications.

12. Table 1 has p values accompanied by PW. What does this stand for…its not defined at the end of the table.

Response: Thank you for your observation. We have removed this comparison as suggested by Reviewer Two.

13. Under results…there is a title “ BMI changes fron baseline to four years” what does this mean? Authors need to generallt improve on the english for clarity of reading.

Response: Thank you for your observation. We have changed the title to improve clarity.

14. In the abstract, its stated that being married was associated with weight gain , however in the result section, its stated that partipants who were not married had a hiogher proportion of being overweight or obese compared to married participants. This is contradicting.

Response: Thank you for your observation. The apparent contradiction arises from distinguishing the unadjusted descriptive statistics presented in Table 3 from the adjusted regression results in Table 4. Table 3 shows the row proportions, indicating that 68.2% of overweight or obese participants were unmarried. This represents a descriptive snapshot of baseline characteristics and does not account for potential confounders. In contrast, Table 4 presents findings from an adjusted multivariable analysis. After controlling for age, waist circumference, CD4 count, and other relevant covariates, being married was independently associated with a 2.34-fold higher risk of developing overweight or obesity (aIRR 2.34; 95% CI: 1.24–4.40). Therefore, the abstract’s conclusion regarding the association between marriage and overweight/obesity is drawn from this adjusted model, which isolates the effect of marital status while accounting for confounding factors.

15. A higher proportion of individuals who were overweight or obese had a higher BMI at baseline compared to those without (22.9 kg/m2 vs. 20.0 kg/m2, p < 0.001). Median waist circumference was higher in the overweight/obese group compared to individuals who were not overweight or obese (79.5 cm vs. 74 cm, p < 0.001). Is this not obvious?

Response: Thanks for the observation. Waist circumference was included in the model as a measure of central adiposity, complementing BMI. It’s a strong independent association with incident overweight/obesity, which highlights the role of visceral fat in early weight-related risk accumulation. Previous studies have demonstrated their role in the development of overweight and obesity. While this association may seem evident, our aim was to confirm it within our population, which has not been previously studied. (1. Sweatt K, Garvey WT, Martins C. Strengths and Limitations of BMI in the Diagnosis of Obesity: What is the Path Forward? Curr Obes Rep. 2024;13: 584–595. doi:10.1007/s13679-024-00580-1).

16. Edit table 3 for clarity

Response: Thank you. We have done edits to make the table clearer.

17. It’s written that Table 4 shows the results on the univariable and multivariable regression analyses of fcators associated with overweight…..this should be a bivariate analysis

Response: Thank you for the observation. We have edited this part in our table.

18. In the multivariable analyis, one year increase in age was significantly associated with a 3% reduced risk of being overweight…where does weight come from given it is not included in the bivariate analysis?

Response: Thank you. The main outcome of the study was overweight/obesity, and hence the interpretation “one-year increase in age was significantly associated with a 3% reduced risk of being overweight/obese”. Age was included in the final model (multivariable analsysis) based on the previous literature as opposed to its significance at bivariable analysis. Hope we were able to get your question. We sincerely appreciate your comment.

19. CD4 counts were also not included in the univariate analysis yet it’s mentioned under the same section on multivariate analysis

Response: Thank you. We did not interpret CD4 in bivariable analysis because it was not statistically significant; however, in the multivariable analysis, it became significant, and we had to interpret it. CD4 was included in the final model (multivariable analysis) based on the previous literature, as opposed to its significance at bivariable analysis

20. Table 4 the upper limit of 95% CI is very close to the null value and so this seems to be a chance finding.

Response: Thank you. We appreciate your insightful comment regarding the upper limit of the 95% confidence interval approaching the null value, which could also indicate a weak association. However, we defined statistical significance as a p-value <0.05 and/or a 95% confidence interval that does not include 1 in the regression model.

Review Comments to the Author

Reviewer #1: 1. The authors should use a consistent nosology for people living with HIV (PLWH); there are two nosologies used eg People living with HIV and people with HIV

Response: Thank you. We have endeavored to use on nosology for people living with HIV throughout the manuscript

2. The findings should be stated and then referenced with figues/table and not the other way round.

Response: Thank you for the observation. We have changed throughout the manuscript.

3. The conclusion did not explicitly answer the objective

Response: Thank you for the observation. We have changed this conclusion to answer the objective

Reviewer #2: The methods and statistical analysis and results sections are not clear. It is not clear how long the follow-up period was, and what the interval of follow-up for each participant was. Also, it is not clear what was done at the different follow-up periods.

Response: Thank you for the observation. We have now clarified what the follow-up period was and the median follow-up period for the participants “This was a longitudinal cohort study with one follow-up assessment per participant. The median follow-up time was 43 months (IQR: 42–44), calculated as the time between baseline enrollment (April 2019–May 2020) and the follow-up visit (December 2022–June 2023)”. We had only two points to collect data (baseline and follow-up period), and we have explicitly said what was done at the two time periods.

It is not clear what the final sample size is, since the authors indicate they recruited 388 participants and included 249, but after removing all the individuals excluded, the remaining number is less than 190, and not 249.

Response: Thank you for the observation. We have now corrected this and justified how we arrived at 249, see Figure 1 and the explanation in the first part of the results section.

The comparison of the baseline characteristics between baseline and follow-up is quite unusual and should be removed.

Response: Thank you for the observation. We have now removed the comparison from Table 1.

The tables need to be edited in line with the journal recommendations.

Response: Thank you. We have edited the tables to reflect journal recommendations

In the write-up on 'univariate analysis, which should be bivariate analysis, he needs to include all the variables that were entered in the multivariate analysis.

Response: Thank you. In the write-up up we only narrated significant variables in bivariable and multivariable analysis, and the selection of the model was based on literature and significance in bivariable analysis.

The authors indicate that after checking for normality, they found data not to be normally distributed, why then do they use the chi-square and the poisson regression for non-normally distributed data?

Response: Thank you for this important methodological question. Chi-square tests were used appropriately for categorical variables (e.g., marital status, education level), which do not require normality. Assumptions of independent observations and adequate expected cell sizes (≥5 in ≥80% of cells) were satisfied. Normality checks (Shapiro-Wilk) were applied only to continuous variables. For these, we used non-parametric Wilcoxon signed-rank tests where appropriate (Table 3). Poisson regression with robust standard errors was used to estimate factors associated with incident overweight/obesity cases. As a generalized linear model, it does not assume normality. We applied robust standard errors to address overdispersion, consistent with recommendations for binary outcomes with >10% incidence. Logistic regression tends to overestimate risk ratios for common outcomes; Poisson regression with robust errors yields more accurate estimates. Continuous predictors (e.g., waist circumference) were included without requiring normality, as the log-link function accommodates skewed distributions.

We would like to thank the reviewers for taking the time to make suggestions that have improved our manuscript. We have extensively revised the manuscript and addressed all concerns and suggestions. We now hope the current manuscript is acceptable for publication.

Please address all correspondence to lukundosiame23@gmail.com . We look forward to hearing from you at your earliest convenience.

Please do not hesitate to contact me should you have further questions.

Yours sincerely,

Dr. Lukundo Siame, Bsc., MBcHB.

Junior Residence Medical Livingstone University Teaching Hospital

---

## [Editor Report · Decision Letter 1]

17 Jul 2025

Dear Dr. Siame,

Thank you for submitting your manuscript to PLOS ONE. After careful consideration, we feel that it has merit but does not fully meet PLOS ONE’s publication criteria as it currently stands. Therefore, we invite you to submit a revised version of the manuscript that addresses the points raised during the review process.

We look forward to receiving your revised manuscript.

Kind regards,

Anne Kapaata

Academic Editor

PLOS ONE

Journal Requirements:

Additional Editor Comments:

Clarifications needed before final approval to publish

While the Author has addressed most queries given in the first rebuttal, it is still unclear with regards to:

1. This appears to be a retrospective cohort study given that data was collected at start and end point. Most ART providing health facilities prescribe 3 monthly ART and clients have to return to the facility every three months for refill. Why did the authors not collect data at least every 6 months for the 3-4 year period and only collect data at baseline and end point?.

2. The total follow-up time for each participants appears to be 3-4 years (between 2019/2020 to 2022-2023), and each participant was followed only once, meaning you collected data from each participant at baseline and at the end of data collection only. We know that weight changes rapidly over short periods of time, and many factors explain these changes. Making conclusions about changes in weight over a four year period may not be very accurate and liable to a lot of biases and confounding.

3. Authors mention in the manuscript that they excluded 63 participants after enrolment due to death, pregnancy, loss to follow-up among others. Please note that in a cohort study, once people are enrolled into the study, they contribute some person time to the follow-up and so should not be excluded especially in the baseline and survival analysis. At what point did these people fall out of the study and how much follow-up time did they contribute to the study. Did any of them have the outcome of interest at the time of dropping out of the study?

4. Table one should not have the follow-up data. It should only contain baseline demographics and should be on 312 participants that were enrolled into the study. Table one still has empty cells in the follow-up section. This is not acceptable.

---

## [Author Response · Author response to Decision Letter 2]

24 Jul 2025

23/07/2025

PLOS ONE Journal

Dear Editor,

Ref: Submission of a revised research article for peer review and publication consideration

Reference to the above-mentioned subject. I am writing to submit a revised original research article titled " Risk Factors for Overweight/Obesity among People Living with HIV on Antiretroviral Therapy: A Cohort Study at a Tertiary Health Facility in Zambia.

We would like to thank the reviewers for taking the time to make suggestions that have improved our manuscript. We have revised the manuscript and addressed all concerns and suggestions. We now hope the current manuscript is acceptable for publication. Below are the point-by-point responses to all comments and suggestions.

Journal Requirements:

Response: thank you, noted.

Response: thank you, all reviewed .

Additional Editor Comments:

Clarifications needed before final approval to publish

While the Author has addressed most queries given in the first rebuttal, it is still unclear with regards to:

1. This appears to be a retrospective cohort study given that data was collected at start and end point.

Response: Thank you for your observation. Indeed, there is a retrospective element to the study. Initial data were collected approximately three years ago, after which the same cohort was recalled, and key study variables were physically collected. Given that some data—such as ART regimen and duration on ART—were obtained from medical records (SmartCare), while other data—such as weight, height, and lipid profiles—were collected directly from participants, we believe this qualifies as an ambidirectional cohort study, which takes into account the retrospective and prospective nature of the data. The manuscript has been revised accordingly.

Most ART providing health facilities prescribe 3 monthly ART and clients have to return to the facility every three months for refill. Why did the authors not collect data at least every 6 months for the 3–4-year period and only collect data at baseline and end point?.

Response: Thank you very much for your concern. While we would have preferred to collect data at six-month intervals as suggested, limited funding constrained us to only two data collection points. In our setting, medical records often have significant gaps, and without a properly designed study, there is a high risk of underpowered results and substantial missing data. Hence we could only recall the participants after securing research funds. Nevertheless, we have acknowledged this limitation in the Discussion section of the manuscript. “The study has some strengths and weaknesses. The study has included only a single follow-up period, making it impossible to determine the exact point when individuals met the criteria for overweight or obesity”

2. The total follow-up time for each participants appears to be 3-4 years (between 2019/2020 to 2022-2023), and each participant was followed only once, meaning you collected data from each participant at baseline and at the end of data collection only. We know that weight changes rapidly over short periods of time, and many factors explain these changes. Making conclusions about changes in weight over a four-year period may not be very accurate and liable to a lot of biases and confounding.

Response: thank you very much. We have revised and modified the discussion and conclusion of this manuscript to be more cautious study.

3. Authors mention in the manuscript that they excluded 63 participants after enrolment due to death, pregnancy, loss to follow-up among others. Please note that in a cohort study, once people are enrolled into the study, they contribute some person time to the follow-up and so should not be excluded especially in the baseline and survival analysis. At what point did these people fall out of the study and how much follow-up time did they contribute to the study. Did any of them have the outcome of interest at the time of dropping out of the study?

Response: Thank you for the observation. Given the study design we used where we had only one follow-up period, we were unable to conduct survival analysis. We did not collect data on the exact time each of the events happened for all the excluded participants; our interest was to analyze data for all the participants who had a measurement on the outcome of interest at follow-up period. Therefore, we had to exclude anyone without the measurement on the outcome of interest. We have included a recommendation in the conclusion “. Longitudinal studies with serial measurements are needed to clarify the impact and timing of weight gain from integrase inhibitors in this population”

4. Table one should not have the follow-up data. It should only contain baseline demographics and should be on 312 participants that were enrolled into the study. Table one still has empty cells in the follow-up section. This is not acceptable.

Response: thank you very much. we have redone Table to only show the baseline characteristics and a supplementary for endeline characteristics of the participant. Regarding including 312 participants; our design in another way is a pre-post study design and our aim was seeing what outcome each participant will end up having at follow-up given the baseline characteristics. Therefore, excluded everyone without the measurement on the outcome of interest (We did not collect data on the exact time each of the events happened for all the excluded participants).

We have revised the manuscript and addressed all concerns raised. We want to thank you all again for the tremendous work and time that you committed to reviewing and correcting our work. Our manuscript is much improved, and we are very grateful.

Please address all correspondence to lukundosiame23@gmail.com . We look forward to hearing from you at your earliest convenience.

Please do not hesitate to contact me should you have further questions.

Yours sincerely,

Dr. Lukundo Siame, Bsc., MBcHB.

Junior Residence Medical Livingstone University Teaching Hospital

---

## [Editor Report · Decision Letter 2]

6 Aug 2025

Risk Factors for Overweight/Obesity among People living with HIV on Antiretroviral Therapy:  An ambidirectional cohort study at a Tertiary Health Facility in Zambia.

PONE-D-25-22448R2

Dear Dr. Lukundo Siame

We’re pleased to inform you that your manuscript has been judged scientifically suitable for publication and will be formally accepted for publication once it meets all outstanding technical requirements.

Kind regards,

Anne Kapaata (PhD)

Academic Editor

PLOS ONE

---

## [Editor Report · Acceptance letter]

PONE-D-25-22448R2

PLOS ONE

Dear Dr. Siame,

I'm pleased to inform you that your manuscript has been deemed suitable for publication in PLOS ONE. Congratulations! Your manuscript is now being handed over to our production team.

Kind regards,

on behalf of

Dr. Anne Kapaata

Academic Editor

PLOS ONE